# Revenue Optimization against Strategic Buyers

**Mehryar Mohri**
Courant Institute of Mathematical Sciences
251 Mercer Street
New York, NY, 10012

**Andrés Muñoz Medina**[*]
Google Research
111 8th Avenue
New York, NY, 10011

## Abstract

We present a revenue optimization algorithm for posted-price auctions when facing a buyer with random valuations who seeks to optimize his $\gamma$-discounted surplus. In order to analyze this problem we introduce the notion of $\epsilon$-strategic buyer, a more natural notion of strategic behavior than what has been considered in the past. We improve upon the previous state-of-the-art and achieve an optimal regret bound in $O(\log T + 1/\log(1/\gamma))$ when the seller selects prices from a finite set and provide a regret bound in $\widetilde{O}(\sqrt{T} + T^{1/4}/\log(1/\gamma))$ when the prices offered are selected out of the interval $[0, 1]$.

## 1 Introduction

Online advertisement is currently the fastest growing form of advertising. This growth has been motivated, among other reasons, by the existence of well defined metrics of effectiveness such as click-through-rate and conversion rates. Moreover, online advertisement enables the design of better targeted campaigns by allowing advertisers to decide which type of consumers should see their advertisement. These advantages have promoted the fast pace development of a large number of advertising platforms. Among them, AdExchanges have increased in popularity in recent years. In contrast to traditional advertising, AdExchanges do not involve contracts between publishers and advertisers. Instead, advertisers are allowed to bid in real-time for the right to display their ad.

An AdExchange works as follows: when a user visits a publisher's website, the publisher sends this information to the AdExchange which runs a second-price auction with reserve (Vickrey, 1961; Milgrom, 2004) among all interested advertisers. Finally, the winner of the auction gets the right to display his ad on the publisher's website and pays the maximum of the second highest bid and the reserve price. In practice, this process is performed in milliseconds, resulting in millions of transactions recorded daily by the AdExchange. Thus, one might expect that the AdExchange could benefit from this information by *learning* how much an advertiser values the right to display his ad and setting an optimal reserve price. This idea has recently motivated research in the learning community on revenue optimization in second-price auctions with reserve (Mohri and Medina, 2014a; Cui et al., 2011; Cesa-Bianchi et al., 2015).

The algorithms proposed by these authors heavily rely on the assumption that the advertisers' bids are drawn i.i.d. from some underlying distribution. However, if an advertiser is aware of the fact that the AdExchange or publisher are using a revenue optimization algorithm, then, most likely, he would adjust his behavior to *trick* the publisher into offering a more beneficial price in the future. Under this scenario, the assumptions of (Mohri and Medina, 2014a) and (Cesa-Bianchi et al., 2015) would be violated. In fact, empirical evidence of strategic behavior by advertisers has been documented by Edelman and Ostrovsky (2007). It is therefore critical to analyze the interactions between publishers and *strategic* advertisers.

---

[*]This work was partially done at the Courant Institute of Mathematical Sciences.

In this paper, we consider the simpler scenario of revenue optimization in posted-price auctions with strategic buyers, first analyzed by Amin et al. (2013). As pointed out by Amin et al. (2013), the study of this simplified problem is truly relevant since a large number of auctions run by AdExchanges consist of only one buyer (or one buyer with a large bid and several buyers with negligible bids). In this scenario, a second-price auction in fact reduces to a posted-price auction where the seller sets a reserve price and the buyer decides to accept it (bid above it) or reject it (bid below).

To analyze the sequential nature of this problem, we can cast it as a repeated game between a buyer and a seller where a strategic buyer seeks to optimize his surplus while the seller seeks to collect the largest possible revenue from the buyer. This can be viewed as an instance of a repeated non-zero sum game with incomplete information, which is a problem that has been well studied in the Economics and Game Theory community (Nachbar, 1997, 2001). However, such previous work has mostly concentrated on the characterization of different types of achievable equilibria as opposed to the design of an algorithm for the seller. Furthermore, the problem we consider admits a particular structure that can be exploited to derive learning algorithms with more favorable guarantees for the specific task of revenue optimization.

The problem can also be viewed as an instance of a multi-armed bandit problem (Auer et al., 2002; Lai and Robbins, 1985), more specifically, a particular type of continuous bandit problem previously studied by Kleinberg and Leighton (2003). Indeed, at every time $t$ the buyer can only observe the revenue of the price he offered and his goal is to find, as fast as possible, the price that would yield the largest expected revenue. Unlike a bandit problem, however, here, the performance of an algorithm cannot be measured in terms of the external regret. Indeed, as observed by Bubeck and Cesa-Bianchi (2012) and Arora et al. (2012), the notion of external regret becomes meaningless when facing an adversary that reacts to the learner's actions. In short, instead of comparing to the best achievable revenue by a fixed price over the sequence of rewards *seen*, one should compare against the simulated sequence of rewards that would have been seen had the seller played a fixed price. This notion of regret is known as *strategic regret* and regret minimization algorithms have been proposed before under different scenarios (Amin et al., 2013, 2014; Mohri and Medina, 2014a). In this paper we provide a regret minimization algorithm for the stochastic scenario, where, at each round, the buyer receives an i.i.d. valuation from an underlying distribution. While this random valuation might seems surprising, it is in fact a standard assumption in the study of auctions (Milgrom and Weber, 1982; Milgrom, 2004; Cole and Roughgarden, 2014). Moreover, in practice, advertisers rarely interact directly with an AdExchange. Instead, several advertisers are part of an ad network and it is that ad network that bids on their behalf. Therefore, the valuation of the ad network is not likely to remain fixed. Our model is also motivated by the fact that the valuation of an advertiser depends on the user visiting the publisher's website. Since these visits can be considered random, it follows that the buyer's valuation is in fact a random variable.

A crucial component of our analysis is the definition of a strategic buyer. We consider a buyer who seeks to optimize his cumulative discounted surplus. However, we show that a buyer who exactly maximizes his surplus must have unlimited computational power, which is not a realistic assumption in practice. Instead, we define the notion of an $\epsilon$-*strategic* buyer who seeks only to approximately optimize his surplus. Our main contribution is to show that, when facing an $\epsilon$-strategic buyer, a seller can achieve $O(\log T)$ regret when the set of possible prices to offer is finite, and an $O(\sqrt{T})$ regret bound when the set of prices is $[0, 1]$. Remarkably, these bounds on the regret match those given by Kleinberg and Leighton (2003) in a truthful scenario where the buyer does not behave strategically.

The rest of this paper is organized as follows. In Section 2, we discuss in more detail related previous work. Next, we define more formally the problem setup (Section 3). In particular, we give a precise definition of the notion of $\epsilon$-strategic buyer (Section 3.2). Our main algorithm for a finite set of prices is described in Section 4, where we also provide a regret analysis. In Section 5, we extend our algorithm to the continuous case where we show that a regret in $O(\sqrt{T})$ can be achieved.

## 2   Previous work

The problem of revenue optimization in auctions goes back to the seminal work of Myerson (1981), who showed that under some regularity assumptions over the distribution $\mathcal{D}$, the revenue optimal, incentive-compatible mechanism is a second-price auction with reserve. This result applies to single-shot auctions where buyers and the seller interact only once and the underlying value distribution is

known to the seller. In practice, however it is not realistic to assume that the seller has access to this distribution. Instead, in cases such as on-line advertisement, the seller interacts with the buyer a large number of times and can therefore infer his behavior from historical data. This fact has motivated the design of several learning algorithms such as that of (Cesa-Bianchi et al., 2015) who proposed a bandit algorithm for revenue optimization in second-price auctions; and the work of (Mohri and Medina, 2014a), who provided learning guarantees and an algorithm for revenue optimization where each auction is associated with a feature vector.

The aforementioned algorithms are formulated under the assumption of buyers bidding in an i.i.d. fashion and do not take into account the fact that buyers can in fact react to the use of revenue optimization algorithms by the seller. This has motivated a series of publications focusing on this particular problem. Bikhchandani and McCardle (2012) analyzed the same problem proposed here when the buyer and seller interact for only two rounds. Kanoria and Nazerzadeh (2014) considered a repeated game of second-price auctions where the seller knows that the value distribution can be either *high*, meaning it is concentrated around high values, or *low*; and his goal is to find out from which distribution the valuations are drawn under the assumption that buyers can behave strategically.

Finally, the scenario considered here was first introduced by Amin et al. (2013) where the authors solve the problem of optimizing revenue against a strategic buyer with a fixed valuation and showed that a buyer can achieve regret in $O\big(\frac{\sqrt{T}}{1-\gamma}\big)$. Mohri and Medina (2014b) later showed that one can in fact achieve a regret in $O(\frac{\log T}{1-\gamma})$ closing the gap with the lower bound to a factor of $\log T$. The scenario of random valuations we consider here was also analyzed by Amin et al. (2013) where an algorithm achieving regret in $O\big(|\mathcal{P}|T^\alpha + \frac{1}{(1-\gamma)^{1/\alpha}} + \frac{1}{\Delta^{1/\alpha}}\big)$ was proposed when prices are offered from a finite set $\mathcal{P}$, with $\Delta = \min_{p \in \mathcal{P}} p^* \mathcal{D}(v > p^*) - p\mathcal{D}(v > p)$ and $\alpha$ a free parameter. Finally, an extension of this algorithm to the contextual setting was presented by the same authors in (Amin et al., 2014) where they provide an algorithm achieving $O\big(\frac{T^{2/3}}{1-\gamma}\big)$ regret.

The algorithms proposed by Amin et al. (2013, 2014) consist of alternating exploration and exploitation. That is, there exist rounds where the seller only tries to estimate the value of the buyer and other rounds where he uses this information to try to extract the largest possible revenue. It is well known in the bandit literature (Dani and Hayes, 2006; Abernethy et al., 2008) that algorithms that ignore information obtained on exploitation rounds tend to be sub-optimal. Indeed, even in a truthful scenario where the UCB algorithm (Auer et al., 2002) achieves regret in $O(\frac{\log T}{\Delta})$, the algorithm proposed by Amin et al. (2013) achieves sub-optimal regret in $O\big(e^{\sqrt{\log T \log \frac{1}{\Delta}}}\big)$ for the optimal choice of $\alpha$ which, incidentally, requires also access to the unknown value $\Delta$.

We propose instead an algorithm inspired by the UCB strategy using exploration and exploitation simultaneously. We show that our algorithm admits a regret that is in $O\big(\frac{\log T}{\Delta} + \frac{|\mathcal{P}|}{\log(1/\gamma)}\big)$, which matches the UCB bound in the truthful scenario and which depends on $\gamma$ only through the additive term $\frac{1}{\log(1/\gamma)} \approx \frac{1}{1-\gamma}$ known to be unavoidable (Amin et al., 2013). Our results cannot be directly compared with those of Amin et al. (2013) since they consider a fully strategic adversary whereas we consider an $\epsilon$-strategic adversary. As we will see in the next section, however, the notion of $\epsilon$-strategic adversary is in fact more natural than that of a buyer who exactly optimizes his discounted surplus. Moreover, it is not hard to show that, when applied to our scenario, perhaps modulo a constant, the algorithm of Amin et al. (2013) cannot achieve a better regret than in the fully strategic adversary.

## 3 Setup

We consider the following scenario, similar to the one introduced by Amin et al. (2013).

### 3.1 Scenario

A buyer and a seller interact for $T$ rounds. At each round $t \in \{1, \ldots, T\}$, the seller attempts to sell some good to the buyer, such as the right to display an ad. The buyer receives a valuation $v_t \in [0, 1]$ which is unknown to the seller and is sampled from a distribution $\mathcal{D}$. The seller offers a price $p_t$,

in response to which the buyer selects an action $a_t \in \{0, 1\}$, with $a_t = 1$ indicating that he accepts the price and $a_t = 0$ otherwise. We will say the buyer *lies* if he accepts the price at time $t$ ($a_t = 1$) while the price offered is above his valuation ($v_t \leq p_t$), or when he rejects the price ($a_t = 0$) while his valuation is above the price offered ($v_t > p_t$).

The seller seeks to optimize his expected revenue over the $T$ rounds of interaction, that is,

$$\mathrm{Rev} = \mathbb{E}\left[\sum_{t=1}^{T} a_t p_t\right].$$

Notice that, when facing a truthful buyer, for any price $p$, the expected revenue of the seller is given by $p\mathcal{D}(v > p)$. Therefore, with knowledge of $\mathcal{D}$, the seller could set all prices $p_t$ to $p^*$, where $p^* \in \mathrm{argmax}_{p \in [0,1]} \, p\mathcal{D}(v > p)$. Since the actions of the buyer do not affect the choice of future prices by the seller, the buyer has no incentive to lie and the seller will obtain an expected revenue of $Tp^*\mathcal{D}(v > p^*)$. It is therefore natural to measure the performance of any revenue optimization algorithm in terms of the following notion of *strategic regret*:

$$\mathrm{Reg}_T = Tp^*\mathcal{D}(v > p^*) - \mathrm{Rev} = \max_{p \in [0,1]} Tp\mathcal{D}(v > p) - \mathbb{E}\left[\sum_{t=1}^{T} a_t p_t\right].$$

The objective of the seller coincides with the one assumed by Kleinberg and Leighton (2003) in the study of repeated interactions with buyers with a random valuation. However, here, we will allow the buyer to behave strategically, which results in a harder problem. Nevertheless, the buyer is not assumed to be fully adversarial as in (Kleinberg and Leighton, 2003). Instead, we will assume, as discussed in detail in the next section, that the buyer seeks to approximately optimize his surplus, which can be viewed as a more natural assumption.

### 3.2 $\epsilon$-strategic Buyers

Here, we define the family of buyers considered throughout this paper. We denote by $x_{1:t} \in \mathbb{R}^t$ the vector $(x_1, \ldots, x_t)$ and define the history of the game up to time $t$ by $H_t := (p_{1:t}, v_{1:t}, a_{1:t})$. Before the first round, the seller decides on an algorithm $\mathcal{A}$ for setting prices and this algorithm is announced to the buyer. The buyer then selects a strategy $\mathcal{B} \colon (H_{t-1}, v_t, p_t) \mapsto a_t$. For any value $\gamma \in (0, 1)$ and strategy $\mathcal{B}$, we define the buyer's discounted expected surplus by

$$\mathrm{Sur}_\gamma(\mathcal{B}) = \mathbb{E}\left[\sum_{t=1}^{T} \gamma^{t-1} a_t (v_t - p_t)\right].$$

A buyer minimizing this discounted surplus wishes to acquire the item as inexpensively as possible, but does not wish to wait too long to obtain a favorable price.

In order to optimize his surplus, a buyer must then solve a non-homogeneous Markov decision process (MDP). Indeed, consider the scenario where at time $t$ the seller offers prices from a distribution $\mathcal{D}_t \in \mathfrak{D}$, where $\mathfrak{D}$ is a family of probability distributions over the interval $[0, 1]$. The seller updates his beliefs as follows: the current distribution $\mathcal{D}_t$ is selected as a function of the distribution at the previous round as well as the history $H_{t-1}$ (which is all the information available to the seller). More formally, we let $f_t \colon (\mathcal{D}_t, H_t) \mapsto \mathcal{D}_{t+1}$ be a transition function for the seller. Let $s_t = (\mathcal{D}_t, H_{t-1}, v_t, p_t)$ denote the state of the environment at time $t$, that is, all the information available at time $t$ to the buyer. Finally, let $S_t(s_t)$ denote the maximum attainable expected surplus of a buyer that is in state $s_t$ at time $t$. It is clear that $S_t$ will satisfy the following Bellman equations:

$$S_t(s_t) = \max_{a_t \in \{0,1\}} \gamma^{t-1} a_t (v_t - p_t) + \mathbb{E}_{(v_{t+1}, p_{t+1}) \sim \mathcal{D} \times f_t(\mathcal{D}_t, H_t)} \left[S_{t+1}(f_t(\mathcal{D}_t, H_t), H_t, v_{t+1}, p_{t+1})\right],$$

(1)

with the boundary condition $S_T(s_T) = \gamma^{T-1}(v_T - p_T)\mathbb{1}_{p_T \leq v_T}$.

**Definition 1.** *A buyer is said to be strategic if his action at time $t$ is a solution of the Bellman equation* (1).

Notice that, depending on the choice of the family $\mathfrak{D}$, the number of states of the MDP solved by a strategic buyer may be infinite. Even for a deterministic algorithm that offers prices from a finite set $\mathcal{P}$, the number of states of this MDP would be in $\Omega(T^{|\mathcal{P}|})$, which quickly becomes intractable. Thus, in view of the prohibitive cost of computing his actions, the model of a fully strategic buyer does not seem to be realistic. We introduce instead the concept of $\epsilon$-strategic buyers.

**Definition 2.** *A buyer is said to be $\epsilon$-strategic if he behaves strategically, except when no sequence of actions can improve upon the future surplus of the truthful sequence by more than $\gamma^{t_0}\epsilon$, or except for the first $0 < t < t_0$ rounds, for some $t_0 \geq 0$ depending only on the seller's algorithm, in which cases he acts truthfully.*

We show in Section 4 that this definition implies the existence of $t_1 > t_0$ such that an $\epsilon$-strategic buyer only solves an MDP over the interval $[t_0, t_1]$ which becomes a tractable problem for $t_1 \ll T$. The parameter $t_0$ used in the definition is introduced to consider the unlikely scenario where a buyer's algorithm deliberately ignores all information observed during the rounds $0 < t < t_0$, in which case it is optimal for the buyer to behave truthfully.

Our definition is motivated by the fact that, for a buyer with bounded computational power, there is no incentive in acting non-truthfully if the gain in surplus over a truthful behavior is negligible.

## 4 Regret Analysis

We now turn our attention to the problem faced by the seller. The seller's goal is to maximize his revenue. When the buyer is truthful, Kleinberg and Leighton (2003) have shown that this problem can be cast as a continuous bandit problem. In that scenario, the strategic regret in fact coincides with the pseudo-regret, which is the quantity commonly minimized in a stochastic bandit setting (Auer et al., 2002; Bubeck and Cesa-Bianchi, 2012). Thus, if the set of possible prices $\mathcal{P}$ is finite, the seller can use the UCB algorithm Auer et al. (2002) to minimize his pseudo-regret.

In the presence of an $\epsilon$-strategic buyer, the rewards are no longer stochastic. Therefore, we need to analyze the regret of a seller in the presence of lies. Let $\mathcal{P}$ denote a finite set of prices offered by the seller. Define $\mu_p = p\mathcal{D}(v > p)$ and $\Delta_p = \mu_{p^*} - \mu_p$. For every price $p \in \mathcal{P}$, define also $T_p(t)$ to be the number of times price $p$ has been offered up to time $t$. We will denote by $T^*$ and $\mu^*$ the corresponding quantities associated with the optimal price $p^*$.

**Lemma 1.** *Let $\mathcal{L}$ denote the number of times a buyer lies. For any $\delta > 0$, the strategic regret of a seller can be bounded as follows:*

$$\text{Reg}_T \leq \mathbb{E}[\mathcal{L}] + \sum_{p\,:\,\Delta_p > \delta} \mathbb{E}[T_p(t)]\Delta_p + T\delta.$$

*Proof.* Let $\mathcal{L}_t$ denote the event that the buyer lies at round $t$, then the expected revenue of a seller is given by

$$\mathbb{E}\left[\sum_{t=1}^{T}\sum_{p\in\mathcal{P}} a_t p_t \mathbb{1}_{p_t=p}(\mathbb{1}_{\mathcal{L}_t} + \mathbb{1}_{\mathcal{L}_t^c})\right] \geq \mathbb{E}\left[\sum_{t=1}^{T}\sum_{p\in\mathcal{P}} a_t p_t \mathbb{1}_{p_t=p}\mathbb{1}_{\mathcal{L}_t^c}\right] = \mathbb{E}\left[\sum_{p\in\mathcal{P}}\sum_{t=1}^{T} \mathbb{1}_{v_t>p}p\mathbb{1}_{p_t=p}\mathbb{1}_{\mathcal{L}_t^c}\right],$$

where the last equality follows from the fact that when the buyer is truthful $a_t = \mathbb{1}_{v_t>p}$. Moreover, using the fact that $\sum_{t=1}^{T}\mathbb{1}_{\mathcal{L}_t} = \mathcal{L}$, we have

$$\mathbb{E}\left[\sum_{p\in\mathcal{P}}\sum_{t=1}^{T} \mathbb{1}_{v_t>p}p\mathbb{1}_{p_t=p}\mathbb{1}_{\mathcal{L}_t^c}\right] = \mathbb{E}\left[\sum_{p\in\mathcal{P}}\sum_{t=1}^{T} \mathbb{1}_{v_t>p}p\mathbb{1}_{p_t=p}\right] - \mathbb{E}\left[\sum_{p\in\mathcal{P}}\sum_{t=1}^{T} \mathbb{1}_{v_t>p}p\mathbb{1}_{p_t=p}\mathbb{1}_{\mathcal{L}_t}\right]$$

$$= \sum_{p\in\mathcal{P}} \mu_p\,\mathbb{E}[T_p(T)] - \mathbb{E}\left[\sum_{t=1}^{T} \mathbb{1}_{v_t>p_t}p_t\mathbb{1}_{\mathcal{L}_t}\right]$$

$$\geq \sum_{p\in\mathcal{P}} \mu_p\,\mathbb{E}[T_p(T)] - \mathbb{E}[\mathcal{L}].$$

Since the regret of offering prices for which $\Delta_p \leq \delta$ is bounded by $T\delta$, it follows that the regret of the seller is bounded by $\mathbb{E}[\mathcal{L}] + \sum_{p\,:\,\Delta_p > \delta} \Delta_p\,\mathbb{E}[T_p(T)] + T\delta$. $\qquad\square$

We now define a *robust UCB* (R-UCB$_L$) algorithm for which we can bound the expectations $\mathbb{E}[T_p(T)]$. For every price $p \in \mathcal{P}$, define

$$\widehat{\mu}_p(t) = \frac{1}{T_p(t)}\sum_{i=1}^{t} p_t \mathbb{1}_{p_t=p}\mathbb{1}_{v_t>p_t}$$

to be the true empirical mean of the reward that a seller would obtain when facing a truthful buyer. Let $L_t(p) = \sum_{i=1}^{t} (a_t - \mathbb{1}_{v_t > p}) \mathbb{1}_{p_t = p} p$ denote the revenue obtained by the seller in rounds where the buyer lied. Notice that $L_t(p)$ can be positive or negative. Finally, let

$$\overline{\mu}_p(t) = \widehat{\mu}_p(t) + \frac{L_t(p)}{T_p(t)}$$

be the empirical mean obtained when offering price $p$ that is observed by the seller. For the definition of our algorithm, we will make use of the following upper confidence bound:

$$B_p(t, L) = \frac{Lp}{T_p(t)} + \sqrt{\frac{2 \log t}{T_p(t)}}.$$

We will use $B^*$ as a shorthand for $B_{p^*}$. Our R-UCB$_L$ algorithm selects the price $p_t$ that maximizes the quantity

$$\max_{p \in \mathcal{P}} \overline{\mu}_p(t) + B_p(t, L).$$

We proceed to bound the expected number of times a sub-optimal price $p$ is offered.

**Proposition 1.** *Let* $P_t(p, L) := \mathbb{P}\big(\big|\frac{L_t(p)}{T_p(t)}\big| + \big|\frac{L_t(p^*)}{T^*(t)}\big| \geq L\big(\frac{p}{T_p(t)} + \frac{p^*}{T^*(t)}\big)\big)$. *Then, the following inequality holds:*

$$\mathbb{E}[T_p(t)] \leq \frac{4Lp}{\Delta_p} + \frac{32 \log T}{\Delta_p^2} + 2 + \sum_{t=1}^{T} P_t(p, L).$$

*Proof.* For any $p$ and $t$ define $\eta_p(t) = \sqrt{\frac{2 \log t}{T_p(t)}}$ and let $\eta^* = \eta_{p^*}$. If at time $t$ price $p \neq p^*$ is offered then

$$\overline{\mu}_p(t) + B_p(t, L) - \overline{\mu}^*(t) - B^*(t, L) \geq 0$$

$$\Leftrightarrow \quad \widehat{\mu}_p(t) + B_p(t, L) + \frac{L_t(p)}{T_p(t)} - \widehat{\mu}^*(t) - B^*(t, L) - \frac{L_t(p^*)}{T^*(t)} \geq 0$$

$$\Leftrightarrow \quad \big[\widehat{\mu}_p(t) - \mu_p - \eta_p(t)\big] + \big[2B_p(t, L) - \Delta_p\big] + \bigg[\frac{L_t(p)}{T_p(t)} - \frac{L_t(p^*)}{T^*(t)} - \frac{Lp}{T_p(t)} - \frac{Lp^*}{T^*(t)}\bigg]$$
$$+ \big[\mu^* - \widehat{\mu}^*(t) - \eta^*(t)\big] \geq 0. \quad (2)$$

Therefore, if price $p$ is selected, then at least one of the four terms in inequality (2) must be positive. Let $u = \frac{4Lp}{\Delta_p} + \frac{32 \log T}{\Delta_p^2}$. Notice that if $T_p(t) > u$ then $2B_p(t, L) - \Delta_p < 0$. Thus, we can write

$$\mathbb{E}[T_p(T)] = \mathbb{E}\bigg[\sum_{t=1}^{T} \mathbb{1}_{p_t = p}(\mathbb{1}_{T_p(t) \leq u} + \mathbb{1}_{T_p(t) > u})\bigg] = u + \sum_{t=u}^{T} \Pr(p_t = p, \, T_p(t) > u).$$

This combined with the positivity of at least one of the four terms in (2) yields:

$$\mathbb{E}[T_p(T)] \leq u + \sum_{t=u}^{T} \Pr\big(\widehat{\mu}_p(t) - \mu_p \geq \eta_p(t)\big) + \Pr\bigg(\frac{L_t(p^*)}{T^*(t)} - \frac{L_t(p)}{T_p(t)} \geq \frac{Lp}{T_p(t)} + \frac{Lp^*}{T^*(t)}\bigg)$$
$$+ \Pr\big(\mu^* - \widehat{\mu}^*(t) > \eta^*(t)\big)$$
$$\leq u + \sum_{t=u}^{T} \Pr\big(\widehat{\mu}_p(t) - \mu_p \geq \eta_p(t)\big) + \Pr\big(\mu^* - \widehat{\mu}^*(t) > \eta^*(t)\big) + P_t(p, L). \quad (3)$$

We can now bound the probabilities appearing in (3) as follows:

$$\Pr\left(\widehat{\mu}_p(t) - \mu_p \geq \sqrt{\frac{2 \log t}{T_p(t)}}\right) \leq \Pr\left(\exists s \in [0, t] \colon \frac{1}{s} \sum_{i=1}^{s} p \mathbb{1}_{v_i > p} - \mu_p \geq \sqrt{\frac{2 \log t}{s}}\right)$$

$$\leq \sum_{s=1}^{t} t^{-4} = t^{-3},$$

where the last inequality follows from an application of Hoeffding's inequality as well as the union bound. A similar argument can be made to bound the other term in (3). Using the definition of $u$ we then have

$$\mathbb{E}[T_p(T)] \leq \frac{4Lp}{\Delta_p} + \frac{32 \log T}{\Delta_p^2} + \sum_{t=u}^{T} 2t^{-3} + \sum_{t=1}^{T} P_t(p, L) \leq \frac{4Lp}{\Delta_p} + \frac{32 \log T}{\Delta_p^2} + 2 + \sum_{t=1}^{T} P_t(p, L),$$

which completes the proof. $\qquad\square$

**Corollary 1.** *Let $\mathcal{L}$ denote the number of times a buyer lies. Then, the strategic regret of R-UCB$_L$ can be bounded as follows:*

$$\mathrm{Reg}_T \leq L\Big(4\sum_{p \in \mathcal{P}} p\Big) + \mathbb{E}[\mathcal{L}] + \sum_{p:\, \Delta_p > \delta} \Big(\frac{32 \log T}{\Delta_p} + 2\Delta_p + \sum_{t=1}^{T} P_t(p, L)\Big) + T\delta.$$

Notice that the choice of parameter $L$ of R-UCB$_L$ is subject to a trade-off: on the one hand, $L$ should be small to minimize the first term of this regret bound; on the other hand, function $P_t(p, L)$ is decreasing in $T$, therefore the term $\sum_{t=1}^{T} P_t(p, L)$ is beneficial for larger values of $L$.

We now show that an $\epsilon$-strategic buyer can only lie a finite number of times, which will imply the existence of an appropriate choice of $L$ for which we can ensure that $P_t(p, L) = 0$, thereby recovering the standard logarithmic regret of UCB.

**Proposition 2.** *If the discounting factor $\gamma$ satisfies $\gamma \leq \gamma_0 < 1$, an $\epsilon$-strategic buyer stops lying after $S = \left\lceil \frac{\log(1/\epsilon(1-\gamma_0))}{\log(1/\gamma_0)} \right\rceil$ rounds.*

*Proof.* After $S$ rounds, for any sequence of actions $a_t$ the surplus that can be achieved by the buyer in the remaining rounds is bounded by

$$\sum_{t=t_0+S}^{T} \mathbb{E}[a_t(v_t - p_t)] \leq \frac{\gamma^{S+t_0} - \gamma^T}{1-\gamma} \leq \frac{\gamma^{S+t_0}}{1-\gamma} \leq \epsilon,$$

for any sequence of actions. Thus, by definition, an $\epsilon$-strategic buyer does not lie after $S$ rounds. $\qquad\square$

**Corollary 2.** *If the discounting factor $\gamma$ satisfies $\gamma \leq \gamma_0 < 1$ and the seller uses the R-UCB$_L$ algorithm with $L = \left\lceil \frac{\log(1/\epsilon(1-\gamma_0))}{\log(1/\gamma_0)} \right\rceil$, then the strategic regret of the seller is bounded by*

$$\left\lceil \frac{\log \frac{1}{\epsilon(1-\gamma_0)}}{\log \frac{1}{\gamma_0}} \right\rceil \Big(4\sum_{p \in \mathcal{P}} p + 1\Big) + \sum_{p:\Delta_p > \delta} \frac{32 \log T}{\Delta_p} + 2\Delta_p + T\delta. \tag{4}$$

*Proof.* Follows trivially from Corollary 1 and the previous proposition, which implies that $P_t(p, L) \equiv 0$. $\qquad\square$

Let us compare our results with those of Amin et al. (2013). The regret bound given in (Amin et al., 2013) is in $O\Big(|\mathcal{P}|T^\alpha + \frac{|\mathcal{P}|^2}{\Delta^{2/\alpha}} + \frac{|\mathcal{P}|^2}{\left(\Delta(1-\gamma_0)\right)^{1/\alpha}}\Big)$, where $\alpha$ is a parameter controlling the fraction of rounds used for exploration and $\Delta = \min_{p \in \mathcal{P}} \Delta_p$. In particular, notice that the dependency of this bound on the cardinality of $\mathcal{P}$ is quadratic instead of linear as in our case. Moreover, the dependency on $\gamma_0$ is in $O(\frac{1}{1-\gamma}^{1/\alpha})$. Therefore, even in a truthful scenario where $\gamma \ll 1$. The dependency on $T$ remains polynomial whereas we recover the standard logarithmic regret. Only when the seller has access to $\Delta$, which is a strong requirement, can he set the optimal value of $\alpha$ to achieve regret in $O\big(e^{\sqrt{\log T \log \frac{1}{\Delta}}}\big)$.

Of course, the algorithm proposed by Amin et al. (2013) assumes that the buyer is fully strategic whereas we only require the buyer to be $\epsilon$-strategic. However, the authors assume that the distribution satisfies a Lipchitz condition which technically allows them to bound the number of lies in the same way as in Proposition 2. Therefore, the regret bound achieved by their algorithm remains the same in our scenario.

## 5 Continuous pricing strategy

Thus far, we have assumed that the prices offered by the buyer are selected out of a discrete set $\mathcal{P}$. In practice, however, the optimal price may not be within $\mathcal{P}$ and therefore the algorithm described in the previous section might accumulate a large regret when compared against the best price in $[0, 1]$. In order to solve this problem, we propose to discretize the interval $[0, 1]$ and run our R-UCB$_L$ algorithm on the resulting discretization. This induces a trade-off since a better discretization implies a larger regret term in (4). To find the optimal size of the discretization we follow the ideas of Kleinberg and Leighton (2003) and consider distributions $\mathcal{D}$ that satisfy the condition that the function $f \colon p \mapsto p\mathcal{D}(v > p)$ admits a unique maximizer $p^*$ such that $f''(p) < 0$.

Throughout this section, we let $K \in \mathbb{N}$ and we consider the following finite set of prices $\mathcal{P}_K = \left\{ \frac{i}{K} \middle| 1 \leq i \leq K \right\} \subset [0, 1]$. We also let $p_K$ be an optimal price in $\mathcal{P}_K$, that is $p_K \in \mathrm{argmax}_{p \in \mathcal{P}_K} f(p)$ and we let $p^* = \mathrm{argmax}_{p \in [0,1]} f(p)$. Finally, we denote by $\Delta_p = f(p_K) - f(p)$ the sub-optimality gap with respect to price $p_K$ and by $\overline{\Delta}_p = f(p^*) - f(p)$ the corresponding gap with respect to $p^*$. The following theorem can be proven following similar ideas to those of Kleinberg and Leighton (2003). We defer its proof to the appendix.

**Theorem 1.** *Let* $K = \left( \frac{T}{\log T} \right)^{1/4}$, *if the discounting factor* $\gamma$ *satisfies* $\gamma \leq \gamma_0 < 1$ *and the seller uses the R-UCB$_L$ algorithm with the set of prices $\mathcal{P}_K$ and $L = \left\lceil \frac{\log(1/\epsilon(1-\gamma_0))}{\log(1/\gamma_0)} \right\rceil$, then the strategic regret of the seller can be bounded as follows:*

$$\max_{p \in [0,1]} f(p) - \mathbb{E}\left[ \sum_{t=1}^{T} a_t p_t \right] \leq C\sqrt{T \log T} + \left\lceil \frac{\log \frac{1}{\epsilon(1-\gamma_0)}}{\log \frac{1}{\gamma_0}} \right\rceil \left[ \left( \frac{T}{\log T} \right)^{1/4} + 1 \right].$$

## 6 Conclusion

We introduced a revenue optimization algorithm for posted-price auctions that is robust against $\epsilon$-strategic buyers. Moreover, we showed that our notion of strategic behavior is more natural than what has been previously studied. Our algorithm benefits from the optimal $O\left( \log T + \frac{1}{1-\gamma} \right)$ regret bound for a finite set of prices and admits regret in $O\left( T^{1/2} + \frac{T^{1/4}}{1-\gamma} \right)$ when the buyer is offered prices in $[0, 1]$, a scenario that had not been considered previously in the literature of revenue optimization against strategic buyers. It is known that a regret in $o(T^{1/2})$ is unattainable even in a truthful setting, but it remains an open problem to verify that the dependency on $\gamma$ cannot be improved. Our algorithm admits a simple analysis and we believe that the idea of making truthful algorithms *robust* is general and can be extended to more complex auction mechanisms such as second-price auctions with reserve.

## 7 Acknowledgments

We thank Afshin Rostamizadeh and Umar Syed for useful discussions about the topic of this paper and the NIPS reviewers for their insightful comments. This work was partly funded by NSF IIS-1117591 and NSF CCF-1535987.

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
