[Supplementary Material]

# A Proof of Theorem 1

**Proposition 3.** *There exist constants $C_1, C_2 > 0$ such that $C_1(p^* - p)^2 < f(p^*) - f(p) < C_2(p^* - p)^2$ for all $p \in [0, 1]$.*

*Proof.* Since the second derivative $f''(p^*)$ exists, it follows that

$$\lim_{p \to p^*} \frac{f(p^*) - f(p)}{(p^* - p)^2} \to \frac{f''(p^*)}{2}.$$

It then follows by definition that there exists $\delta$ such that $0 < |p - p^*| < \delta$ implies $0 < -\frac{f''(p^*)}{4}(p^* - p)^2 \leq f(p^*) - f(p) \leq -\frac{3f''(p^*)}{4}(p^* - p)^2$. On the other hand, the continuity of $g \colon p \to \frac{f(p^*) - f(p)}{p^* - p}$ on the set $X = \{p \in [0, 1] | |p^* - p| \geq \delta\}$, as well as the compactness of $X$ implies

$$0 < \min_{p \in X} g(p) \leq g(p) \leq \max_{p \in X} g(p),$$

where we have used the fact that $g(p) > 0$ for all $p \in X$. The result of the proposition straightforwardly follows from these observations. □

**Proposition 4.** *Let $p_i = \frac{i}{K} \in \mathcal{P}_K$ and define $\overline{\Delta}_i = \Delta_{p_i}$. Then, $\overline{\Delta}_i > C_1\left(p^* - \frac{i}{K}\right)^2$ for all $i$. Moreover, there exists a reordering $i_0, \ldots, i_{K-1}$ such that $\overline{\Delta}_{i_j} \geq C_1\left(\frac{j}{2K}\right)^2$.*

*Proof.* From Proposition 3 it follows that $\Delta_i > C_1\left(p^* - \frac{i}{K}\right)^2$. The reordering is defined recursively as follows $i_0 = \operatorname{argmin}_i \left(p^* - \frac{i}{K}\right)^2$ and $i_j = \operatorname{argmin}_{i \notin \{i_0, \ldots, i_{j-1}\}} \left(p^* - \frac{i}{K}\right)^2$. Since there are at most $j - 1$ elements in thes set $\mathcal{P}_K$ at distance at most $\frac{j-1}{K}$ from $p^*$, it follows that $\Delta_{i_j} \geq \left(p^* - \frac{i_j}{K}\right)^2 \geq \left(\frac{j}{2K}\right)^2$. □

**Proposition 5.** *The optimal price $p_K$ satisfies, $f(p_K) \geq f(p^*) - \frac{C_2}{K^2}$.*

*Proof.* Let $\widehat{p}$ be the element in $\mathcal{P}_K$ closer to $p^*$, then $(\widehat{p} - p^*)^2 \leq {\frac{1}{K}}^2$ and by Proposition 3 we have

$$\frac{C_2}{K^2} \geq f(p^*) - f(\widehat{p}) \geq f(p^*) - f(p_K).$$

□

**Theorem 1.** *Let $K = \left(\frac{T}{\log T}\right)^{1/4}$, if the discounting factor $\gamma$ satisfies $\gamma \leq \gamma_0 < 1$ and the seller uses the R-UCB$_L$ algorithm with set of prices $\mathcal{P}_K$ and $L = \lceil \frac{\log(1/\epsilon(1-\gamma_0))}{\log(1/\gamma_0)} \rceil$, then the strategic regret of the seller can be bounded as follows:*

$$\max_{p \in [0,1]} f(p) - \mathbb{E}\left[\sum_{t=1}^{T} a_t p_t\right] \leq C\sqrt{T \log T} + \left\lceil \frac{\log \frac{1}{\epsilon(1-\gamma_0)}}{\log \frac{1}{\gamma_0}} \right\rceil \left(\left(\frac{T}{\log T}\right)^{1/4} + 1\right).$$

*Proof.* If $p_K = \operatorname{argmax}_{p \in \mathcal{P}_K} f(p)$, then by Proposition 5 we have:

$$\max_{p \in [0,1]} Tf(p) - \mathbb{E}\left[\sum_{t=1}^{T} a_t p_t\right] = \max_{p \in [0,1]} Tf(p) - Tf(p_K) + Tf(p_K) - \mathbb{E}\left[\sum_{t=1}^{T} a_t p_t\right]$$

$$\leq \frac{TC_2}{K^2} + Tf(p_K) - \mathbb{E}\left[\sum_{t=1}^{T} a_t p_t\right]$$

$$= C_2\sqrt{T \log T} + Tf(p_K) - \mathbb{E}\left[\sum_{t=1}^{T} a_t p_t\right]. \tag{5}$$

The last term in the previous expression corresponds to the regret of the buyer when using a discrete set of prices. By Corollary 2 this term can be bounded by

$$\left\lceil \frac{\log \frac{1}{\epsilon(1-\gamma_0)}}{\log \frac{1}{\gamma_0}} \right\rceil \left( 4 \sum_{p:\,\Delta_p > \delta} p + 1 \right) + \sum_{p:\,\Delta_p > \delta} \frac{32 \log T}{\Delta_p} + 2\Delta_p + T\delta.$$

Letting $i_j$ be as in Proposition 4, we have

$$\Delta_{p_{i_j}} = \overline{\Delta}_{i_j} + f(p_K) - f(p^*) \geq C_1 \left( \frac{j}{2K} \right)^2 - \frac{C_2}{K^2}.$$

Letting $\delta = \sqrt{\frac{\log T}{T}}$ and $j_0 = \sqrt{\frac{8C_2}{C_1}}$ we have $\Delta_{p_{i_j}} \geq \frac{C_1 j^2}{8K^2}$ for $j \geq j_0$. Therefore

$$\sum_{p:\,\Delta_p > \delta} \frac{32 \log T}{\Delta_p} + 2\Delta_p + T\delta \leq 32 \log T \left( \frac{j_0}{\delta} + \frac{8K^2}{C_1} \sum_{j=j_0}^{K} \frac{1}{j^2} \right) + 2K + \frac{T\sqrt{\log T}}{\sqrt{T}}$$

$$\leq 32 \log T \left( \sqrt{\frac{8C_2 T}{C_1 \log T}} + \frac{4\pi^2}{3C_1} \frac{\sqrt{T}}{\sqrt{\log T}} \right) + 2 \left( \frac{T}{\log T} \right)^{1/4} + \sqrt{T \log T}$$

$$= \left( 32 \left( \sqrt{\frac{8C_2}{C_1}} + \frac{4\pi^2}{3C_1} \right) + 1 \right) \sqrt{T \log T} + 2 \left( \frac{T}{\log T} \right)^{1/4}.$$

Finally, we have $\sum_{p:\,\Delta_p > \delta} p \leq K + 1 = \left( \frac{T}{\log T} \right)^{1/4} + 1$. Substituting these bounds into (5) gives the desired result. $\qquad\square$