[Reviews · NeurIPS 2015]

Submitted by Assigned_Reviewer_1

This paper considers the problem of a seller selling an item to a single strategic buyer over a period of T rounds. As in a classic bandit problem, the seller attempts to learn the optimal posted price for the item by running an explore/exploit problem over the set of possible prices. However the buyer, knowing that the seller is learning about his value distribution, may lie in the hopes that the seller will eventually settle on lower prices. This problem has seen some interest recently but the models tend to differ slightly. The three defining features of the model in this paper are:

1. Buyers realized a value from their value distribution at each time step t. (as opposed to having a fixed value over all rounds).

2. They discount their utility over time (this is actually common to all models, and a necessary component to any low-regret scenario).

3. The buyer acts epsilon-strategically, meaning that if the truthful strategy (telling the truth for the remaining time period) guarantees a payoff within epsilon of the optimal strategy at any point, the buyer will use the truthful strategy.

Under this model, the authors show that a modification to the the standard UCB algorithm, denote by UCB_L where L is a free parameter to be set later, and show that it achieves the best known regret within this class of models (although similar work has a slightly different model, so the comparison is not direct). Furthermore it achieves the optimal regret when the buyer acts truthfully.

This is a nice paper that contributes to an interesting and compelling problem within the intersection of machine learning and game theory. The logic of the paper's main proof, which is a clever adaptation of the standard UCB proof, is well-written and easy to follow. The main result is not ground-breaking but it is a solid contribution to the study of online revenue optimization against strategic buyers. The result is perhaps a little incremental, in the sense that the main result is similar to other work but with a twist to the model, but I think the study of this problem is nascent enough to warrant such a contribution.

The one comment I have is that the author's definition of epsilon-strategic buyers could be more explicit. I admit that I first only skimmed their definition because I assumed it was a simple adaptation of the notion of epsilon-best response (from a standard game theory model) but it's not. I could imagine other readers who are familiar with concepts like epsilon Nash equilibria also making the same mistake. I think the definition used in this paper is a fine one but it just might need to be spelled out a bit more.

A few remarks:

- The statement of lemma 1 reads "... for delta >0, the strategic regret of a buyer can be bounded...". I assume this should be the strategic regret of seller. If I misunderstood and it in fact should be the strategic regret of the buyer, please let me know.

- On page 7, the set of inequalities that begins on like ~334: In the second inequality, the indicator variable reads "1_(v_i > p_2)". I assume the 2 in p_2 is a typo and should be just p.

Summary: This is a nice paper that contributes to an interesting and compelling problem within the intersection of machine learning and game theory.

Submitted by Assigned_Reviewer_2

The authors proposed a robust algorithm for the seller to sell items to a single \epsilon-strategic buyer in multiple rounds. The main contribution is a (somewhat tight) theoretical bound on seller's regret.

Pros. Even though it is not the first paper on online revenue maximization for strategic buyers, it seems to be the first paper considering agents that choose a strategy that is not worst than the optimal strategy by more than a given \epsilon. This setting seems to be original. The proof is done by bounding the number of rounds that an \epsilon-strategic buyer may lie under the proposed R-UCB mechanism, which sheds some light on the structure of the problem. The paper is very well-written.

Cons.

On the other hand, the paper is not earthshaking. The mechanism is based on the existing UCB mechanism. Another concern is that I am not totally convinced by the logic in Section 4 in arguing that \epsilon-strategic agents are more natural (not that I am against this claim). Section 4 shows that in order for an agent to compute the optimal strategy, she is facing an intractable MDP. However, I didn't see discussion on the complexity for an \epsilon-strategic to compute a strategy that is no more than \epsilon away from the optimal solution, and moreover, the strategy that minimizes the number of lies among all such strategies. Maybe I missed an obvious point.
Summary: While the paper is not extremely novel, it does provide some interesting results and insight into a new, and potentially realistic, setting.

Submitted by Assigned_Reviewer_3

What part of the analysis breaks down if the buyers simply try to maximize their surplus?

If computing an optimal strategy for a buyer is computationally infeasible, how did previous work that did not assume e-strategic buyers deal with this problem?

I have not checked the proofs in detail but if everything checks out, this is pretty good paper.

Summary: The authors propose a revenue optimization for posted price auctions when the buyers are strategic. They assume buyers want to maximize their discounted surplus. They show a logarithmic regret bound when the prices the seller can pick from are finite.

Submitted by Assigned_Reviewer_4

The paper introduce the realistic assumption of the strategic buyers. The first analysis on the finite price set assumption showed an optimal regret bound.

The proof is correct. This paper is well written, and the literature review is sufficient. Related work is well discussed.
Summary: This paper presents a revenue optimization algorithm based on UCB algorithm.

Submitted by Assigned_Reviewer_5

I wasn't convinced by a number of assumptions in the paper.

First, the authors make a combination of assumptions (discounting, epsilon-strategic, and most unreasonably, that buyers minimizing the number of lies among eps-best responses) that near-trivially reduces the buyers to almost-truthful buyers (except for a bounded number of iterations); the truthful case is already well understood.

I also think the combination of a single strategic buyer with a fresh i.i.d. valuation at every time step is not well motivated.

(As opposed to the already-solved fixed valuation case, which makes sense.)

The previous papers that the authors cite that consider new i.i.d. valuations each time step are thinking of bidders being drawn anew from a large population at each time step (so any one bidder has a fixed valuation, but a random bidder shows up each time step).

Note that strategic issues are not as relevant in such large population models.
Summary: Light review.

See comments below.

Author Feedback
Author rebuttal: We thank all reviewers for their comments.

Reviewer 1:
we thank the reviewer for his positive feedback. We will further discuss and clarify the definition of an epsilon-strategic buyer in the final version. We will also fix typos or other minor issues mentioned by the reviewer.

Reviewer 2:
we view the simplicity of our algorithm and the fact that it is an extension of UCB as positive features. Note that it was not clear that such an extension was possible: the results of [2] tended to suggest the opposite. Also, we have taken note of the reviewer's concerns and will further elaborate on the motivation and explanation of epsilon-strategic buyers in the final version of our paper.

Reviewer 3:
we thank the reviewer for his feedback.

Reviewer 4:
some of the assumptions adopted such as discounting are common. We will further elaborate on the motivation for epsilon-strategic buyers. Our assumptions are definitely not making the problem trivial (see comments to Reviewer 2)

Regarding the stochasticity assumption, as mentioned in the paper, advertisers in an AdExchange do not always bid directly on the AdExchange. Instead, it is common for an ad network to bid on their behalf. In this scenario, the buyer becomes the ad network and receives at each time step a new value (the value of the highest bidder in his network at the time), which can be thought of as being drawn from a distribution. That distribution can be straightforwardly estimated by the network but certainly not by the exchange. Another motivation for the assumption considered is that the inventory of an AdExchange changes at any time, e.g., if the ad is displayed to different users, the value of the advertiser for a particular ad slot changes too.

Reviewer 5:
we agree with the reviewer and our current research work is focused on extending these results to general mechanisms.

Reviewer 6:
- "What part of the analysis breaks down if the buyers simply try to maximize their surplus?" The bound on the number of lies cannot be enforced.

- "If computing an optimal strategy for a buyer is computationally infeasible, how did previous work that did not assume epsilon-strategic buyers deal with this problem?" For the fixed valuation setting, the optimal strategy of the buyer reduces to making the seller believe that she has a different *fixed* valuation. Therefore, simulating a game between a buyer and a seller is simple (see [1]). For a stochastic setting, the previous work did not address this problem at all.

[1] Mehryar Mohri and Andres Munoz Medina. Optimal Regret Minimization in Posted-Price Auctions with Strategic Buyers, NIPS 2014.
[2] Amin, K., Rostamizadeh, A., Syed, U. Learning Prices for Repeated Auctions with Strategic Buyers. NIPS 2013.